# Comprehensive approaches for assessing extinction risk of endangered tropical pitcher plant *Nepenthes talangensis*

**Angga Yudaputra**[1]*, **Inggit Puji Astuti**[1], **Tri Handayani**[1], **Hartutiningsih Siregar**[1], **Iyan Robiansyah**[1], **Sri Wahyuni**[1], **Arief Noor Rachmadiyanto**[1], **Danang Wahyu Purnomo**[1], **Vandra Kurniawan**[1], **Yupi Isnaini**[1], **Frisca Damayanti**[1], **Rizmoon Nurul Zulkarnaen**[1], **Joko Ridho Witono**[2], **Izu Andry Fijridiyanto**[2], **Yuzammi**[2], **Arief Hidayat**[2], **Mustaid Siregar**[3], **Esti Munawaroh**[3], **Fitriany Amalia Wardhani**[4], **Puguh Dwi Raharjo**[5], **Ana Widiana**[6], **Wendell P. Cropper Jr**[7]

1 Research Center for Plant Conservation, Botanic Gardens, and Forestry–National Research and Innovation Agency (BRIN), Bogor, Indonesia, 2 Research Center for Biosystematics and Evolution–National Research and Innovation Agency (BRIN), Bogor, Indonesia, 3 Research Center for Ecology and Ethnobiology–National Research and Innovation Agency (BRIN), Bogor, Indonesia, 4 Research Center for Limnology and Water Resources–National Research and Innovation Agency (BRIN), Bogor, Indonesia, 5 Research Center for Geological Resources–National Research and Innovation Agency (BRIN), Bogor, Indonesia, 6 Faculty of Science and Technology, UIN Sunan Gunung Djati Bandung, Bandung, Indonesia, 7 School of Forest, Fisheries, and Geomatics Sciences, University of Florida, Gainesville, FL United States of America

* angg022@brin.go.id

**Data Availability Statement:** All relevant data are within the manuscript and its Supporting Information files.

## Abstract

It has been 23 years since the conservation status of highland tropical pitcher plant *Nepenthes talangensis* was assessed in 2000. A number of existing threats (anthropogenic and environmental) may be increasing the risk of extinction for the species. A better understanding of the ecology and conservation needs of the species is required to manage the wild populations. Specifically, better information related to population distributions, ecological requirements, priority conservation areas, the impact of future climate on suitable habitat, and current population structure is needed to properly assess extinction risks. A better understanding of the requirements of the species in its natural habitat would benefit for successfully securing the species at Botanic Gardens. We have identified 14 new occurrence records of *N. talangensis* in Mount Talang. Study on the ecological requirement using Random Forest (RF) and Artificial Neural Network (ANN) suggested that elevation, canopy cover, soil pH, and slope are four important variables. The population of *N. talangensis* was dominated by juvenile and mature (sterile) individuals, we found only a few mature males (7 individuals) and females (4 individuals) in the sampled areas. Our modelling of current conditions predicted that there were 1,076 ha of suitable habitat to very highly suitable habitat in Mount Talang, which is 14.7% of the total area. Those predicted habitats ranged in elevation from 1,740–2,558 m. Suitable habitat in 2100 was predicted to decrease in extent and be at higher elevation in the less extreme climate change scenario (SSP 1–2.6) and extreme climate change scenario (SSP 5–8.5). We projected larger habitat loss in the SSP 5–8.5 compared to the SSP 1–2.6 climate change scenario.. We proposed the category CR B1ab(iii,

**Funding:** This study was funded by Nagao Natural Environment Foundation (NEF). The funders had no role in study design, data collection and analysis, decision to publish, or preparation of the manuscript. The URL: https://www.nagaofoundation.or.jp/e/

**Competing interests:** The authors have declared that no competing interests exist.

v), C2a(ii) as the new conservation status of *N. talangensis*. The status is a higher category of threat compared to the current status of the species (EN C2b, ver 2.3). *Nepenthes talangensis* seedlings and cuttings established in a Botanic Garden have relatively high survival rate at about 83.4%. Sixty percent of the seeds germinated in growth media successfully grew to become seedlings.

## Introduction

*Nepenthes talangensis* Nerz & Wistuba is an endemic pitcher plant with a restricted distribution range, only found in Mount Talang, West Sumatra [1]. It is a highland pitcher plant and grows in mossy forests and upper montane forests at 1,800 to 2,500 m elevation near the summit of Mount Talang [2, 3]. It commonly grows as rosettes and climbs in low shrubs [4]. In an early study, Bunnemeijer's specimen collection from Mount Talang was determined to be *N. bongso* [5]. *Nepenthes talangensis* has been thought to be closely related to *N. bongso* [6, 7]. However, recent studies confirmed that both are distinct species based on morphological characters [3, 8]. *Nepenthes talangenis* has two types of pitchers (lower pitcher and upper pitcher). These two types of pitchers have different colours and shapes. The upper pitcher is relatively slender and longer than lower pitcher [8]. *Nepenthes talangensis* has a relatively large peristome that maximizes the effective trapping surface [9].

According to the last assessment [1], it is classified as endangered with very limited information. It has been about 23 years since the last assessment so there may be significant changes in populations or species distribution. Some disturbances may lead the decline of its population in the wild, for instance, habitat fragmentation due to the existence of hiking trails and illegal harvesting by plant enthusiasts [10]. Past studies have characterized the taxonomic, systematic, physiological, and ecological characteristics of the species [4, 8, 11–13].

Due to continuing threats to the population and habitat, and also the limited available information about *N. talangensis*, it is important to carry out a comprehensive study of the ecological requirements, current population status, and in-situ and ex-situ conservation strategies. In order to address these problems, our study includes the following objectives: 1). New occurrence data and ecological requirement modelling, 2). Estimating the current population size structure, 3). Prioritizing conservation areas, 4). Recognizing the trend of future suitable habitats under future climate scenarios, 5). Assessing the conservation status based on IUCN Red List, and 6). Understanding the survival rate of the species during securing efforts in a Botanic Garden.

## Materials and methods

### Study site

Surveys of the endangered tropical pitcher plant *Nepenthes talangensis* were carried out in Mount Talang, an active stratovolcano located in Solok, West Sumatra. It has two crater lakes with a summit of 2,597 m [14, 15]. The lowest daily average low temperature of Mount Talang is 18˚C, and the highest temperature hits to 26˚C [16]. This area receives an average rainfall of 3,000 mm/year and has steep slopes of more than 45% [17]. The study area covers around 7,345.8 ha.

### *Nepenthes talangensis* surveys and characterizing ecological requirements

The data collection was carried out by surveying the accessible locations on Mount Talang. These locations were accessed from three different hiking trails. The ecological and population data within a 20 x 20 m$^2$ plot size were measured at each location where the species was present. The coordinate points where species presence were recorded relative to the nearest hiking trail. The Quantum GIS Desktop 3.18.1 software was used to map the occurrence records [18].

Physical environment variables included elevation, slope, aspect, soil moisture, soil pH, litter thickness, and canopy cover. The relationships of these environmental variables to habitat quality were evaluated using Random Forest (RF) [19] and Artificial Neural Network (ANN) [20] models. The analyses used R packages "randomForest" [21] and "neuralnet" [22]. R studio version 1.2.5042 was used to run the models [23]. Soil samples were analysed to quantify soil chemical composition, soil pH, base saturation, and cation exchange capacity. Four "R packages" were used to do PCA analysis, those were "devtools" [24], "doParallels" [25], "ggplot2" [26], and "ggbiplot" [27]. PCA was used to show a clustering of samples based on their similarity. There were two PCA axes: PCA 1 (x-axis) was the first principal direction and PCA 2 (y-axis) was the second most important direction. Plant association were defined as the plant species found inside the plots. A species composition similarity index was calculated using the jaccard index [28, 29] and visualisation was presented using a 'heat map' [30].

### Population size structure

Population structure data was derived by grouping the individuals into: seedling, juvenile, mature sterile, mature (female) and mature (male). The population size data were obtained by counting all individuals within each plot. The life stages of *N. talangensis* was categorized into three classes (seedlings, juveniles and mature plants) based on their stem lengths and reproductive status. Seedlings had stem lengths < 10 cm, juveniles were 10–20 cm, mature sterile were > 20 cm (without inflorescences), mature male were > 20 cm (with inflorescences), and mature female were > 20 cm (with inflorescences) [31].

### Prioritizing conservation areas

In order to develop a tool for prioritizing conservation areas for *Nepenthes talangenis*, spatial modelling incorporating habitat characteristics where the species was present and occurrence records were used as inputs to the model. Climatic data with a 30 arc second (∼ 1 km) spatial resolution was extracted from WorldClim version 2.1 climate data for 1970–2000. These variables including annual mean temperature, precipitation of wettest month and precipitation of driest month [32]. Four soil variables include soil type, soil pH, soil organic carbon, and cation exchange capacity with a 250 m spatial resolution were derived from SoilGrids—global gridded soil information [33]. Land cover was also applied as a predictor of habitat suitability. Land cover data were derived from Peta Rupabumi Digital Indonesia [34]. Topography data was extracted from NASA Shuttle Radar Topography Mission (SRTM), it was available at a 0.27-arc second resolution [35]. These environmental layers and occurrence records were used as inputs to the model. An ensemble model [36] constructed by combining the prediction of Random Forest (RF) and Support Vector Machine (SVM), was used to predict the suitable habitat of the species. The R "dismo" and R "sdm" package were used to generate the model prediction [37]. Two evaluation metrics: Area Under Curve (AUC) and True Skill Statistics (TSS) were used to validate the performance of model prediction. The value of AUC in range 0.9–1 (excellent), 0.8–0.9 (good), 0.7–0.8 (fair), 0.6–0.7 (poor), and 0.5–0.6 (fail) [38]. Whereas, TSS statistics can be interpreted as the following: values < 0.4 were poor, 0.4–0.8 useful, and > 0.8 good to excellent [39].

Finally, the result of the prediction map was further analysed to identify high priority conservation areas. The predictive map was classified into five classes: unsuitable habitat (0–0.2), barely suitable habitat (0.2–0.4), suitable habitat (0.4–0.6), highly suitable habitat (0.6–0.7), and most suitable habitat (0.7–1.0) [40, 41]. After grouping the potential habitats, the predicted area and elevation were calculated for each class of potential habitat.

### Future suitable areas under future climate scenario

Future climate scenario variables were derived from WorldClim version 2.1 [32]. Global Climate Model (GCM) MIROC 6 with Shared Socioeconomic Pathways (SSP 1–2.6 and SSP 5–8.5) was used to model future habitat suitability for *N. talangensis*. The future climate data was available at a 30 arc second (∼1 km) spatial resolution. These future climate variables were combined with other variables (topography, soil, land cover) that were used as inputs of the model. The predictive map represented the future suitable distribution of *N. talangensis* in the year 2100.

### Conservation status assessment

We assessed the extinction risk of *Nepenthes talangenis* using the IUCN Red List categories and criteria version 3.1 [42]. Due to data availability constraints, we only used criteria B (geographic range), C (small population size and decline), and D (small and restricted population) to assess and classify the species into one of the following categories: Critically Endangered (CR), Endangered (EN), Vulnerable (VU), Near Threatened (NT), and Least Concern (LC). The extent of occurrence (EOO) and area of occupancy (AOO) of the species used in criterion B were calculated using Geospatial Conservation Assessment Tool (GeoCAT) [43], whilst the population size value used in criteria C and D was estimated based on the total number of mature individuals.

### Securing individuals at Botanic Gardens

Several individuals of *N. talangensis* were collected from Mount Talang. Seedlings, cuttings and fruits were taken from the natural habitat in order to preserve the species. Three types of growth media (roasted husks, moss, and mixed media consist of cocopeat + roasted husks + moss) were used to test establishment. Seedlings and cuttings were planted in the growth media, while seeds were grown in several combinations of growth media through a plant tissue culture technique. The survival rates of seedlings and cuttings were observed during establishment periods in the Cibodas Botanic Gardens. The germination of seeds and establishment of seedling and cuttings were conducted for 0–20 weeks of observation.

## Results

### *Nepenthes talangensis* surveys and characterizing ecological requirements

We found *Nepenthes talangensis* in 14 previously undocumented locations during our field surveys in Mount Talang (Fig 1). Those locations ranged from 1,819 to 2,489 m elevation. Machine learning models may use many inputs with complex relationships to each other and with the model outputs. A variety of techniques can be used to evaluate the relative importance of the inputs. The Mean Decrease Gini metric integrates the variable importance estimates over multiple trees with multiple splits. In the Random Forest (RF) model elevation was associated with the highest value (most important) of the model Mean Decrease Gini (4.673) and litter thickness had the lowest value (0). The four variables most important for predicting habitat suitability are elevation, canopy cover, slope and soil pH (Fig 2A). In the Artificial Neural

**Fig 1. Occurrence records of *Nepenthes talangensis* in Mount Talang, West Sumatra, Indonesia.**

Network (ANN) model elevation had the highest importance value (<30) and aspect was the lowest value (less than 5). There were four variables in the ANN (elevation, canopy cover, soil pH, and soil moisture) that had high importance values. Elevation, soil moisture, slope and litter thickness were positively associated with the species. Three input variables (canopy cover,

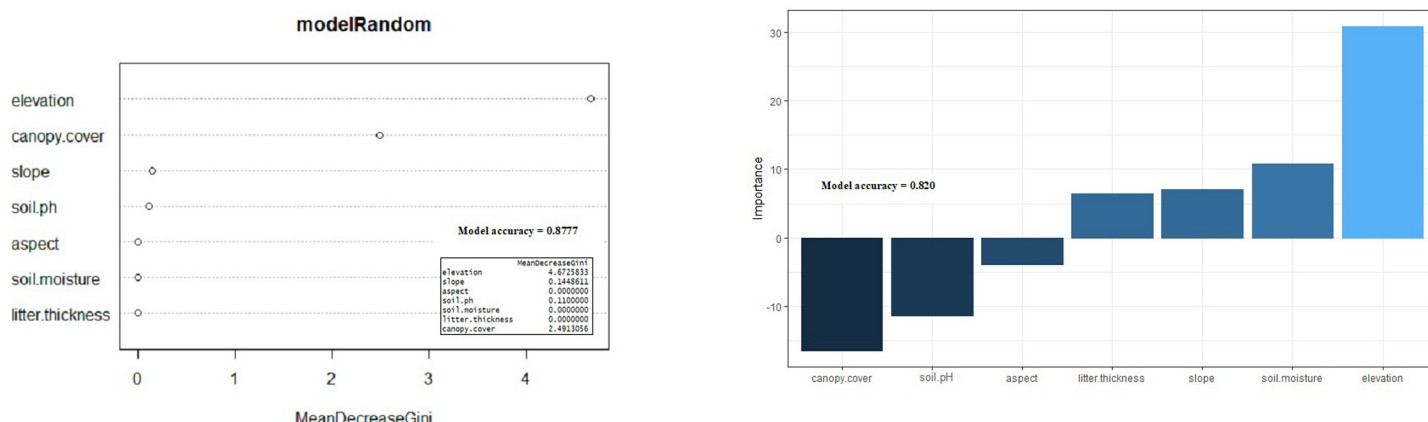

**Fig 2.** The importance variables that influence the occurrence of *Nepenthes talangensis*: a) Random Forest (RF), b). Artificial Neural Network (ANN).

soil pH, and aspect) have negative importance values which means those variables negatively influence the species (Fig 2B). The RF has a model accuracy of 0.877 and the ANN has a model accuracy of 0.820. Both the RF and ANN found elevation, canopy cover and soil pH to be the most important variables in driving habitat suitability (Fig 2A and 2b).

The first principal component explained 76.4% of the variability in the data. Variable correlation plots graphically depict the relationship between all of the variables. Variables were grouped together based on their correlations. Several variables ($AI^{3+}$, N total, CEC, $K_2O$ potential, C-organic, CEC, $P_2O_5$ potential, $Na^{2+}$, $H^+$) were positively correlated, while other variables (C/N ratio, pH $H_2O$, pH KCL) were negatively correlated (Fig 3A and 3B).

Species composition is potentially an important attribute of plant communities, but high variability and diversity can make it difficult to interpret. A heat map of species composition similarity helps to understand the ecological patterns. This heat map has an X axis representing plots where the species was present and the Y axis represents the plant species present inside the plots. In the heat map visualization (Fig 4), there were three distinct groups. A darker grid colour indicates a high abundance, and a brighter grid colour indicates a low abundance. Plots 11, 12, 13, and 14 have similar species compositions, dominated by *Medinilla*, *Cyathea*, *Elaeocarpus* and mosses, plots 1, 2, 3, 4, 5, and 6 were dominated by *Vaccinium*, *Pandanus*, and *Anaphalis*, and plots 7, 8, 9, and 10 were characterized by *Parkia*, *Lasianthus*, *Ardisia*, *Passiflora* and *Lithocarpus* (Fig 4). An average Jaccard similarity index among plots 11, 12, 13, and 14 was > 0.8, for plots 1, 2, 3, 4, 5, and plot 6 was > 0.6, and for plots 7, 8, 9, and plot 10 was > 0.7.

## Population size structure

The juvenile growth stage has the largest number individuals (78 individuals), followed by mature (69 individuals) and seedlings (54 individuals). The largest size class of seedlings was stem lengths from 2 to 4 cm (Fig 5A). Individuals with stem lengths from 16 to18 cm dominated the juvenile growth stage (Fig 5B) and individuals with stem lengths from 20 to 50 cm (the smallest mature size class) were more numerous than other mature size classes (Fig 5C).

## Prioritizing conservation areas

Visualization of potential locations with suitable habitat for protection, and perhaps establishment of new populations of an endangered species can be an effective aid for conservation planning. The model predicted relative habitat suitability (in a range of 0.0 to 1.0) that we grouped into 5 categories. Unsuitable habitat had values of 0–0.2, barely suitable habitat had values of 0.2–0.4 suitable habitat had a range of 0.4–0.6, highly suitable habitat had values of 0.6–0.7, and the most suitable predicted habitat ranged from 0.7–1.0 (Fig 6). The area classified into the unsuitable habitat category had the largest area (5,381.3 ha or 73%) at the elevation range from 812 to 1,605 m, barely suitable habitat (0.2–0.4) included an area of 888 ha (12.1%) from 1,605 to 1,740 m elevation, suitable habitat (0.4–0.6) had an area of 496.4 ha (6.8%) from 1,740 to 1,917 m, highly suitable habitat (0.6–0.7) had the smallest area of 212.4 ha (2.9%) from 1,917 to 2,080 m, and the most suitable habitat class (0.7–1.0) had an area of 367.7 ha (5%) from 2,080–2,558 m (Fig 7). This predictive model is a relatively good predictor of habitat quality with an AUC of 0.97 and a TSS of 0.95. Values of AUC and TSS greater than 0.90, is typical of models that effectively distinguish between presence and absence in observed locations.

Climate variables are important predictors of habitat suitability, allowing predictions of change in habitat suitability associated with climate change scenarios. Predictive future habitat quality of *N. talangensis* using the SSP 1–2.6 scenario showed that class (0–0.2) covered

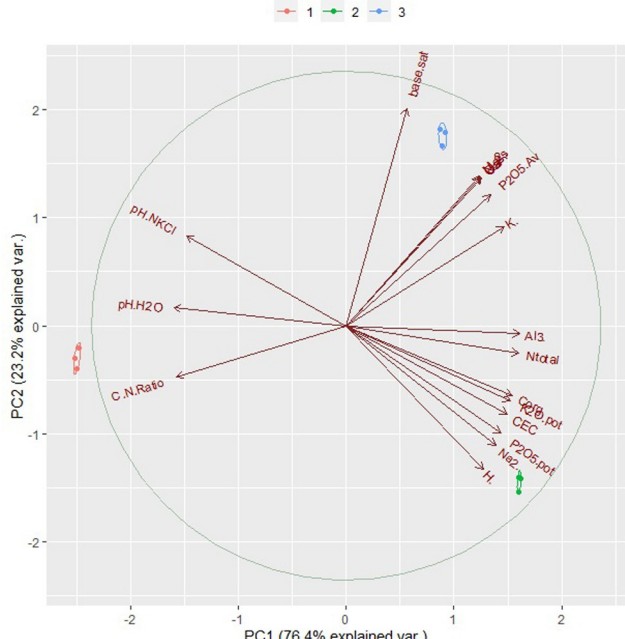
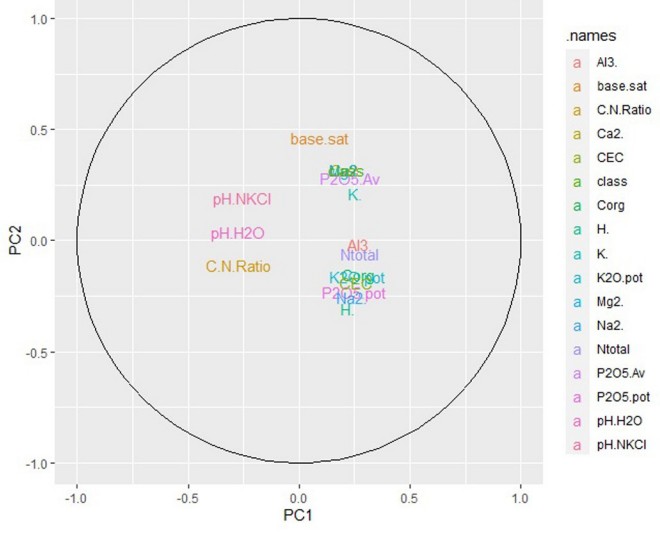

**Fig 3.** Principal Component Analysis (PCA) of soil components in different locations: a). PCA Biplot, b) PCA circle plot.

6,191.62 ha (80.4%) at the elevation range between 812 to 1,820 m, class (0.2–0.4) had an area of 552.69 ha (7.2%) at 1,820 to 2,090 m elevation, class (0.4–0.6) had an area of 351.02 ha (4.6%) at 2,090 to 2,315 m, class (0.6–0.7) had an area 151.29 ha (2%) at 2,315 to 2,356 m, and class (0.7–1.0) had an area of 456.41 ha (5.9%) at 2,356–2,558 m (Fig 8). The predicted future suitable habitat of *N. talangensis* using the SSP 5–8.5 scenario showed that class (0–0.2) covered 6,500.98 ha (84.6%) at an elevation range between 812 to 1,914 m, class (0.2–0.4) had an area of 501.61 ha (6.5%) at 1,914 to 2,051 m elevation, class (0.4–0.6) had an area of 364.01 ha (4.7%) at 2,051 to 2,253 m, class (0.6–0.7) had an area 170.84 ha (2.2%) at 2,253 to 2,400 m, and class (0.7–1.0) had an area 149.05 ha (1.9%) at 2,400–2,558 m (Fig 9).

## Conservation status assessment

GeoCAT showed that the species had an EOO and AOO of 12 km$^2$. Since the species had an EOO<100 km$^2$, in a single location, and inferred to experience continuing decline in area, extent and quality of habitat as well as in the number of mature individuals, under criterion B it could be categorized as CR B1ab(iii,v). Our surveys located a total of 69 mature individuals of the species. Therefore, the species qualified for the category of CR under criterion C2a(ii), i.e. number of mature individuals<250, inferred to experience continuing decline, and all the mature individuals were in one population. For criterion D, the species could be assessed as EN since its number of mature individuals was higher than 50 but less than 250.

## Securing *Nepenthes talangensis* in Botanic Gardens

Only a few seedlings and cuttings were taken from Mount Talang. From six seedlings that were grown in the greenhouse, five remained alive during the five months of observation (Fig 10). From five cuttings, only four were still alive as characterised by some new leaf emergence. During the five months of the germination experiment, there were 20 seedlings that germinated from the seeds and 17 of those developed a pitcher in the roasted husks growth media,

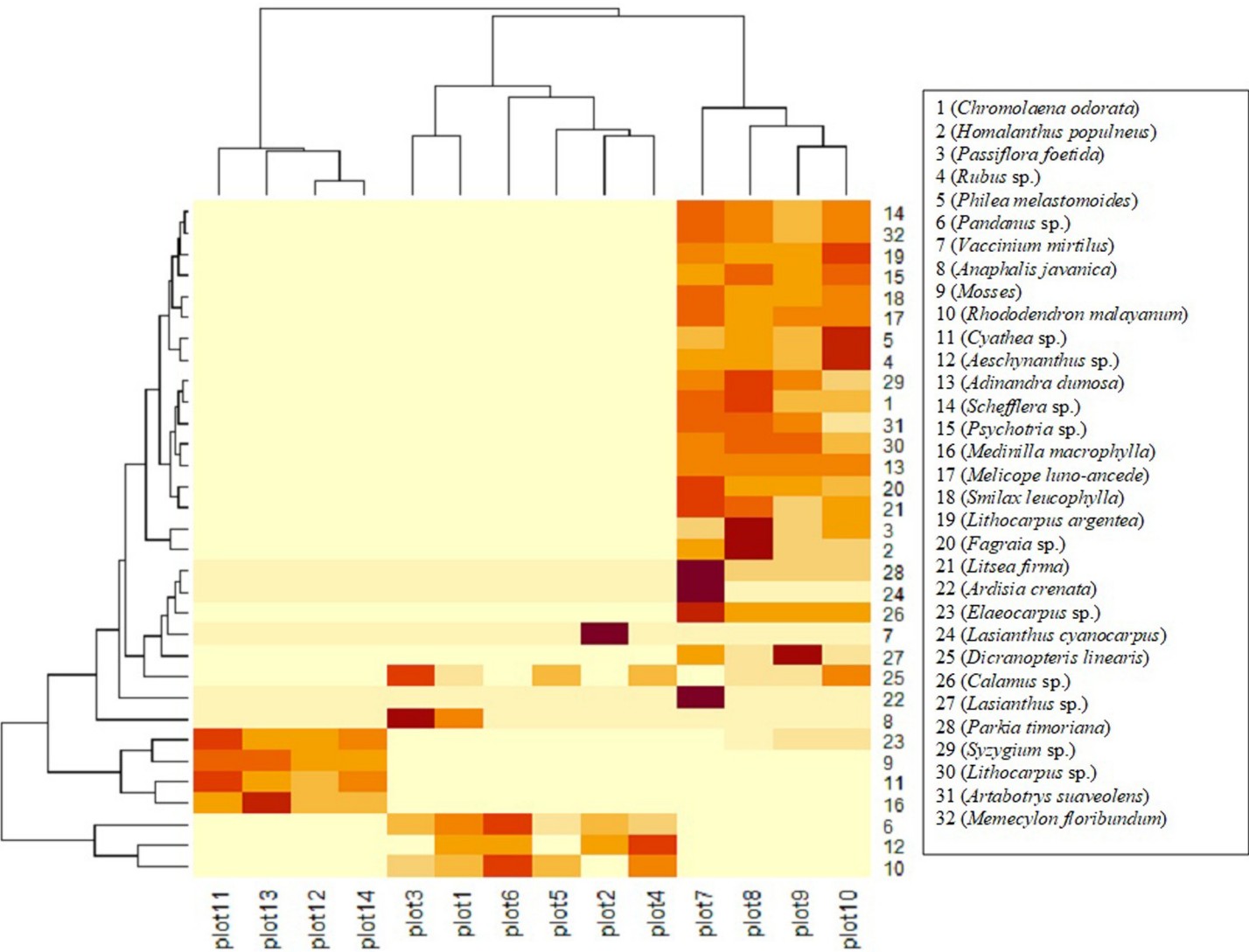

**Fig 4. Heat map generated from species composition data in several plots observation.**

18 seedlings (14 with a pitcher) in the moss media, 13 seedlings (seven with a pitcher) in the cocopeat media, and 18 seedlings (14 with a pitcher) in the mixed media.

## Discussion

### *Nepenthes talangensis* surveys and characterizing ecological requirements

*Nepenthes talangensis* is a rare species with only a few occurrence records found during the field surveys, mostly in upper elevations near the summit of Mount Talang. Some locations could not be sampled due to the topography. The environmental characteristics were measured or determined for each presence location. Characterizing the habitat variables at presence locations is valuable in terms of understanding the ecological requirements of *N. talangensis*. Both machine learning models Random Forest (RF) and Artificial Neural Network (ANN)) predict that similar environmental variables are associated with presence in the wild. These variables are elevation, canopy cover, slope and soil pH. Elevation is identified as the

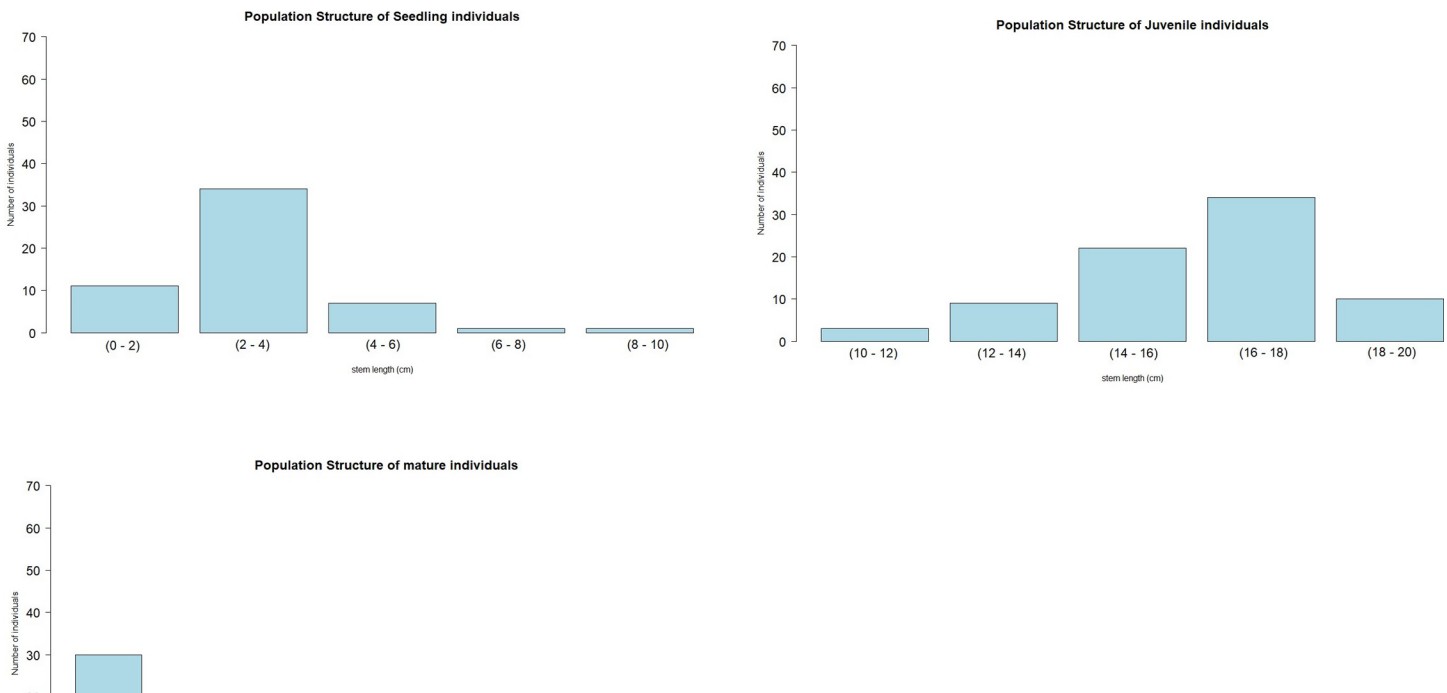

**Fig 5.** Population size and structure of *Nepenthes talangensis* at different growth stage: a). Seedling individuals, b). Juvenile individuals, c). Mature individuals.

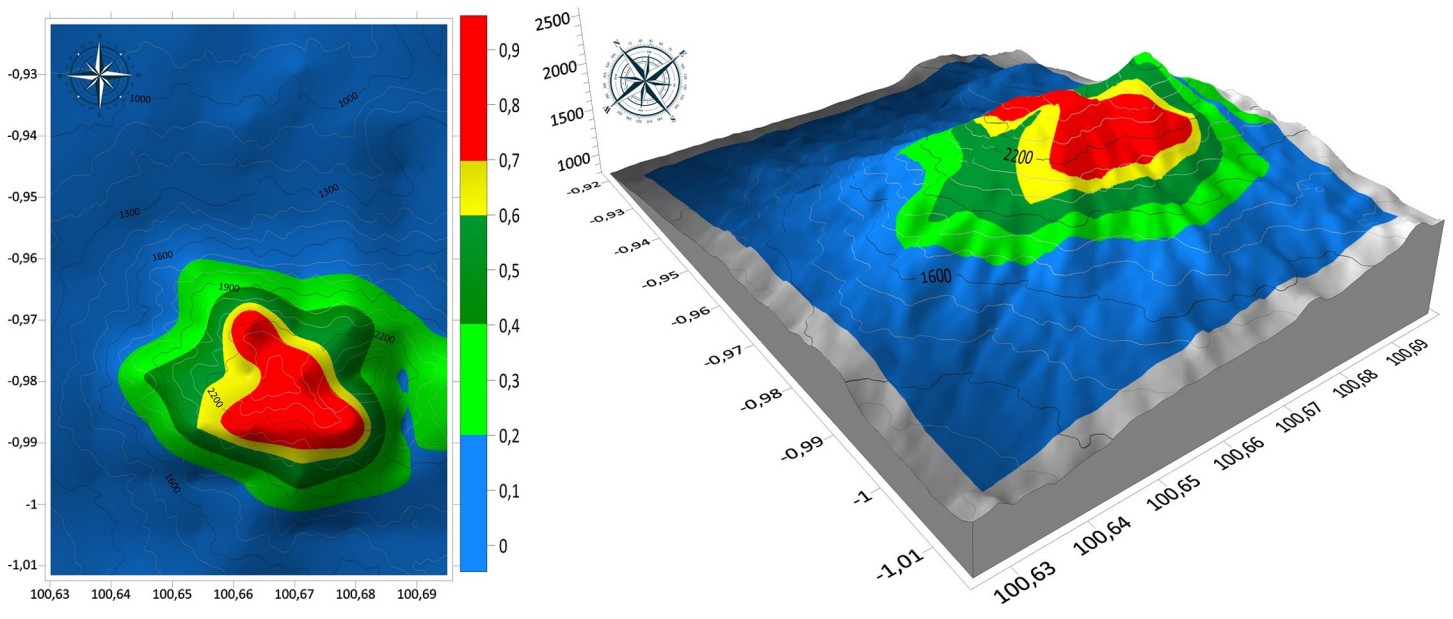

**Fig 6. Predicted suitable habitat map of *Nepenthes talangensis* in Mount Talang.**

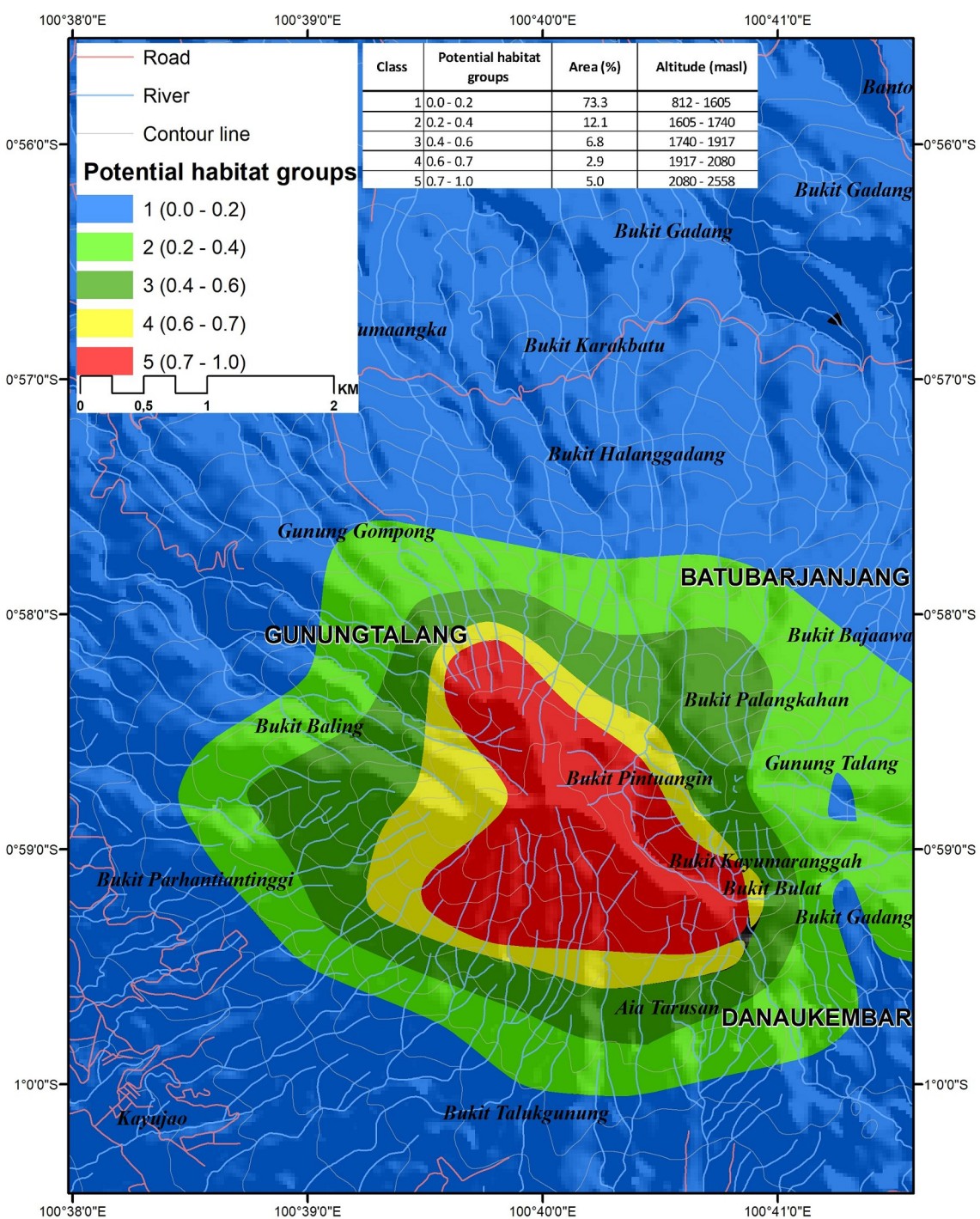

| Class | Potential habitat groups | Area (%) | Altitude (masl) |
|---|---|---|---|
| 1 | 0.0 - 0.2 | 73.3 | 812 - 1605 |
| 2 | 0.2 - 0.4 | 12.1 | 1605 - 1740 |
| 3 | 0.4 - 0.6 | 6.8 | 1740 - 1917 |
| 4 | 0.6 - 0.7 | 2.9 | 1917 - 2080 |
| 5 | 0.7 - 1.0 | 5.0 | 2080 - 2558 |

**Fig 7. The predicted areas of potential habitats and predicted elevation range for each habitat class.**

most important variable determining the species presence in the wild. In the field surveys, the species is seen from 1,819 to 2,489 m elevation. It is found in three different areas: upper montane forest from 1,819 m to 2,360 m, mossy forest with 2,489 m elevation at the summit of Mount Talang, and steep hillsides at 2,239 to 2,288 m elevation near the summit. High elevation is characterized by severe environmental conditions that drastically change diurnally [44].

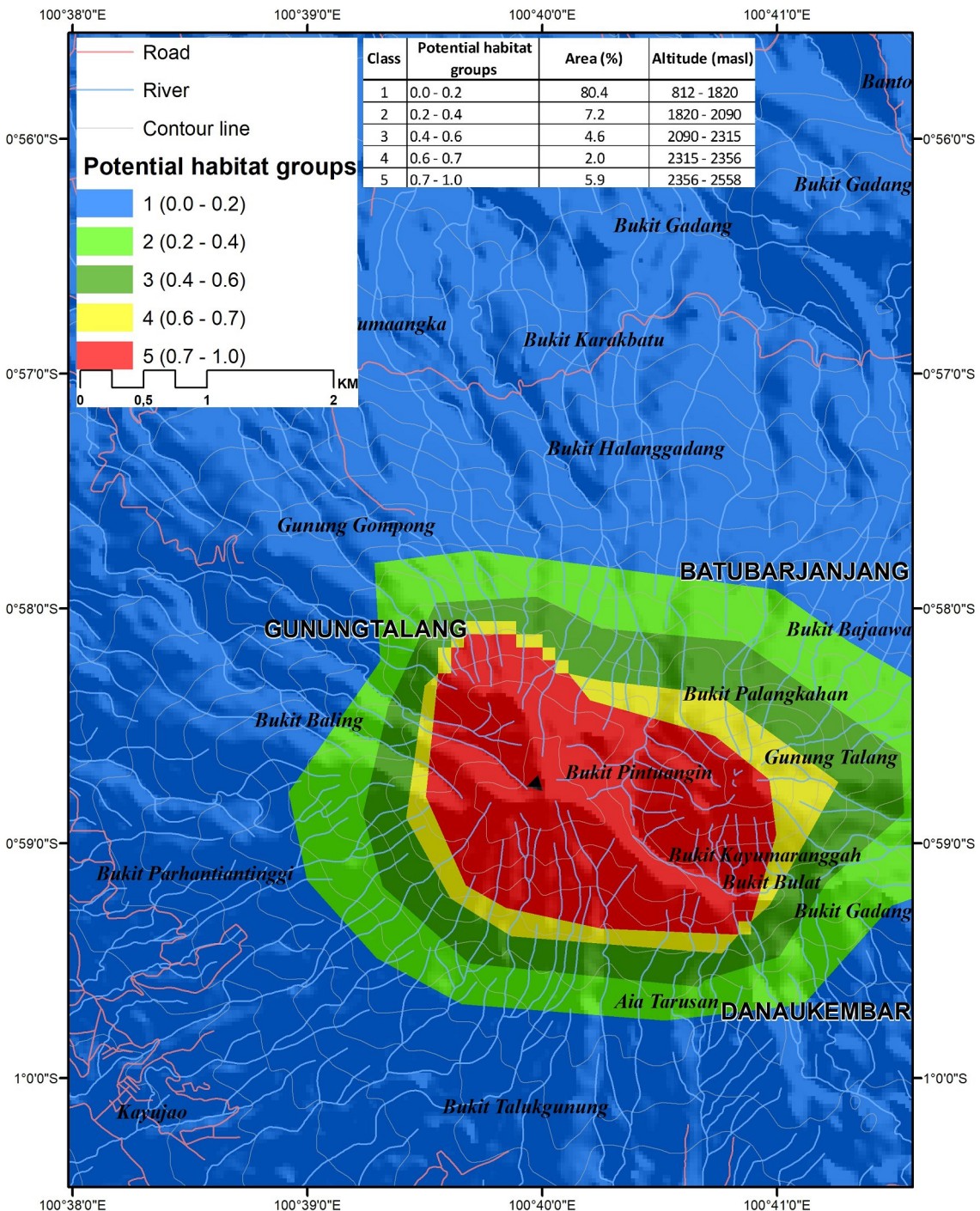

**Fig 8. The predicted areas and predicted elevation range of its future potential habitats in 2100 using GCM MIROC 6 scenario SSP 1–2.6.**

Temperature tends to decrease 0.5 °C per 100 m of increasing elevation [45] and rainfall tends to decrease upslope in many tropical mountains [46]. These conditions can be less favourable for some plants [47]. However, *N. talangensis* is able to adapt and survive in the higher elevations. A previous study found that highland *Nepenthes* have different anatomical structures

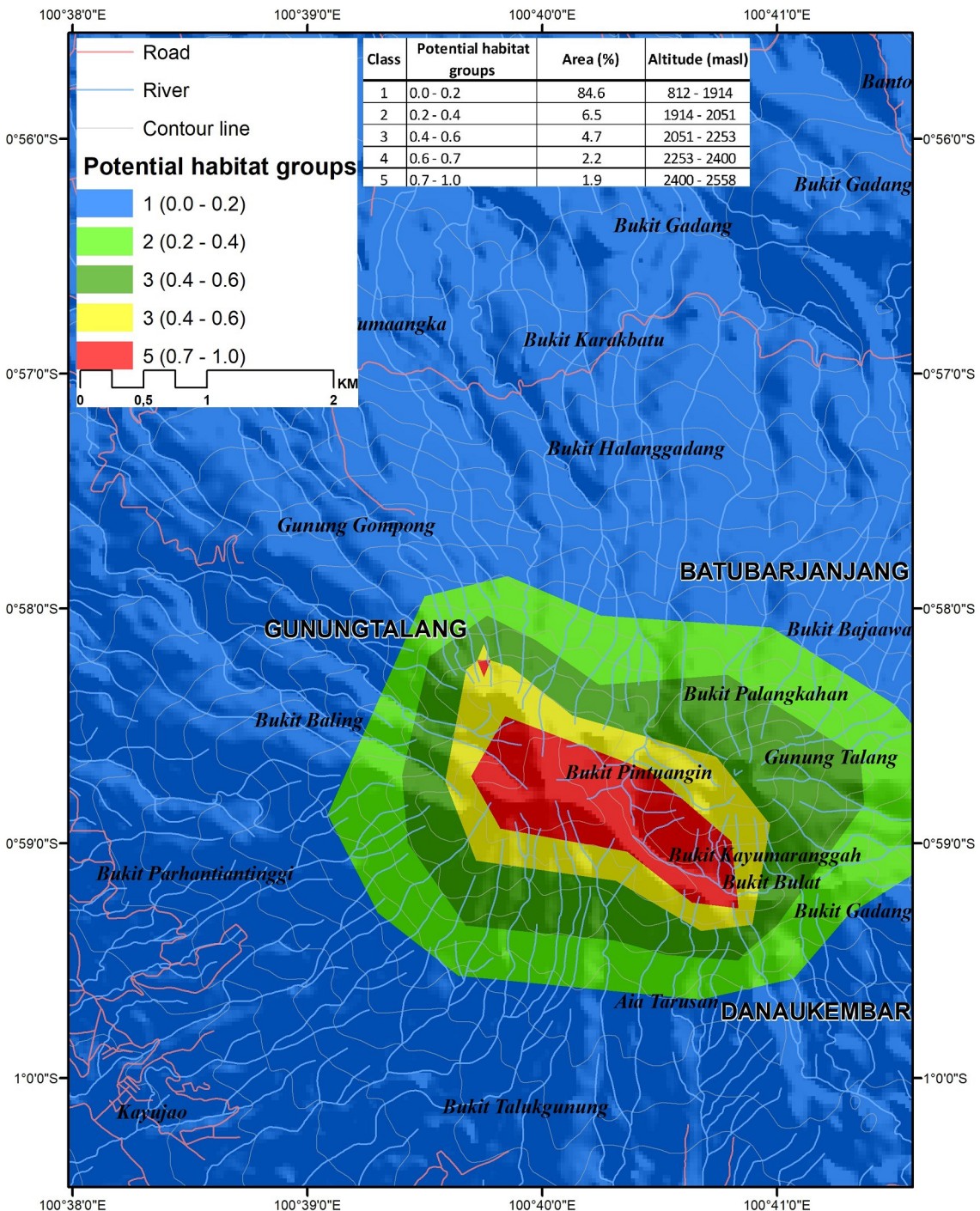

**Fig 9. The predicted areas and predicted elevation range of its future potential habitats in 2100 using GCM MIROC 6 scenario SSP 5–8.5.**

(thicker leaf, larger hypodermis, thicker cuticle, smaller leaf) compared to lowland *Nepenthes* associated with the level of environmental variability [48]. Canopy cover is another important variable affecting the species, the locations where the species was found have canopy cover between 0 and 67.68%. The locations on exposed steep slopes near the summit have 0–5%

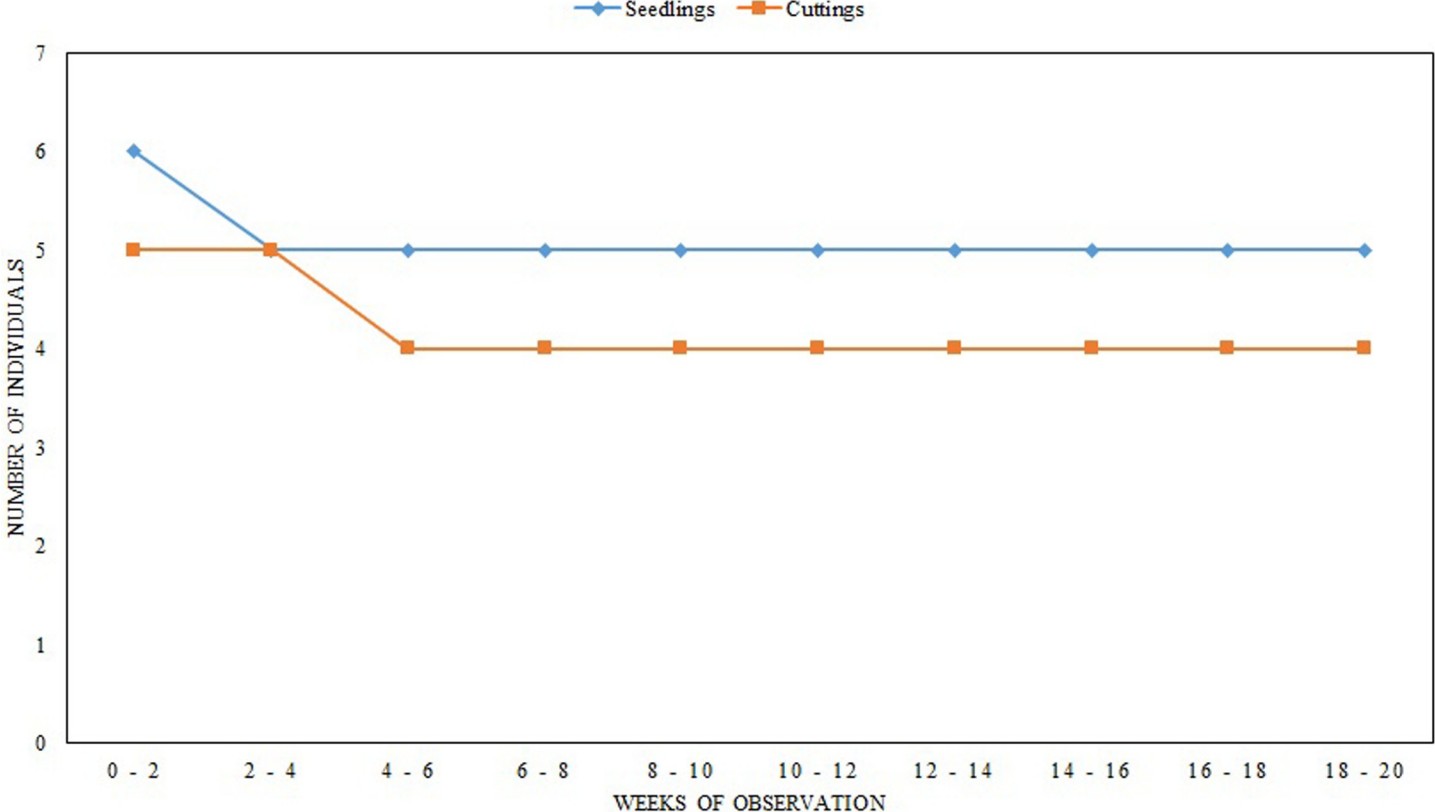

**Fig 10. Survival rate of seedlings and cuttings of *Nepenthes talangensis* in greenhouse of Cibodas Botanic Gardens.**

canopy cover, locations on sunny areas in mossy forests have 42.05–54.16% canopy cover, and locations in upper montane forests have 53.5–67.68% canopy cover. Some individuals grow well on exposed areas on steep slopes near mountain summits by climbing on short vegetation or growing on the ground. Even though some individuals have a relatively short appearance, they often have many pitchers. The pitcher has a distinct colour with the mouth of pitcher typically coloured red to darker red. Our results are consistent with a previous study that found that *N. mirabilis* produces numerous pitchers on exposed habitat in the York Peninsula in Australia [49].

In mossy forests, some individuals can be found in the open short vegetation and some of them are epiphytes climbing on tall trees. The branches and trunks of trees in mossy forests are covered with mosses. Those individuals also have numerous pitchers, but the mouth of pitcher tends to have a brighter yellow colour. Individuals growing in upper montane forests tend to have a denser canopy cover, some individuals climb tall trees with a length of more than 3 m. It seems that the individuals climbing the tall trees to gain more sunlight. The individuals growing at the upper montane region have numerous and larger pitchers, but some of them appear dry and have died. Few dead pitchers are found fallen to the ground from tall canopies. The denser the canopy, the higher the species climbs up the standing trees. Canopy cover closely correlates to the amount of light received by the plants and light intensity affects the growth rate and pitcher production in *Nepenthes* [50]. Commonly, *Nepenthes* requires full sunlight to grow well, and fails to produce flowers under heavy dense canopies [51]. Slope is another environmental variable affecting the distribution of *N. talangensis*. Individuals are

typically found in steep slopes (30–45%). The species can grow well from mid slope to hill slope. Although mid and hill slope usually have lower soil moisture compared to bottom slope, those locations are suitable for *N. talangensis*. In these cases, the soil surface is covered by mosses with soils losing less moisture due to evaporation. Meanwhile, slope in mossy forests and upper montane forests is varied from 0–25%. Locations with *N. talangensis* present have soil pH values ranging from 5.4 to 6.4, categorized as acidic soil. This finding similar to several previous studies that found that *Nepenthes mirabilis* grows well in soil pH ranging between 5 and 6.1 [52], *Nepenthes* (*Nepenthes ampullaria*, *Nepenthes gracilis* and *Nepenthes rafflesiana*) grow in soil pH values ranging from 3.74 to 4.21 [53], *Nepenthes gracilis* grows in soil pH values of 4.17 to 5.56 [54], and three other *Nepenthes* species (*N. gracilis*, *N. rafflesiana* and *N. ampularia*) are found in very acidic soil with pH values of 3.60 to 3.95 [55]. The majority of carnivorous plants are found in sites with poor nutrients, and sunny and moist conditions during the growing season [56, 57].

The surveyed plots can be grouped into three different locations (steep hills near the summit, mossy forests near the mountain summit, and upper montane forests). These locations have different plant compositions where *N. talangensis* are found. The locations on steep slopes are dominated by *Pandanus* sp., mosses that covering the hill's surface, *N. inermis* grows side by side with *N. talangensis* and short stature vegetation (*Vaccinium* and *Rhododendron*). In mossy forest is dominated by Moss, *Medinilla*, *Cyathea*, and *N. gymnamphora*. Locations in upper montane forests are dominated by tall vegetation (*Memecylon*, *Lithocarpus*) and *N. bongso* grows close to *N. talangensis*. Several previous studies reported that *N. mirabilis* and *N. rafflesiana* are found in Padang Alan Forest dominated by *Shorea albida* and Padang Kerumutan Forest dominated by *Combretocarpus rotundatus* [51]. *Nepenthes reinwardtiana* grows epiphytically on *Dipterocarpus oblongifolius* trees [51]. In addition, *N. hookeriana* is found climbing among *Gleichenia bushes* [58]. It is likely *Nepenthes* tends to use any standing trees to climb if it grows under dense canopy, but it creeps on the soil surface and climbs short vegetation in sunny and exposed areas.

Soil fertility on the steep slopes near the summit is relatively low based on N total, $P_2O_5$ available, $K_2O$ potential and CEC measurements. The low fertility soils generally have a large loamy sand component. The sand fraction has poor physical properties because the particles are easily separated due to weak cohesive forces between the particles [59], so the ability to bind water [60] and essential elements becomes low [61]. The low cohesive forces in the sand fraction will also increase the rate of evaporation, thus accelerating the loss of water and nutrients [62]. However, the organic carbon content in the locations where *N. talangensis* grows on rocky hills is relatively high due to the large amount of litter that has completely decomposed. So even though organic carbon is high, litter nutrients only slowly available to be absorbed by plants. *Nepenthes* is able to live in conditions of low soil fertility such as on rocky slopes because these plants are adapted to sites with low nitrogen availability [3, 63, 64]. The soil surface in mossy forests is dominated by moss litter. This litter is characterized by a very high organic C content (48.03%), so *N. talangensis* in this location does not grow on true soil but can grow on the litter resulting from the decomposition of moss biomass. The quality of the litter as for growth at this location can be categorized as very high with respect to the total N, available $P_2O_5$, potential $K_2O$ and CEC. The soil pH surrounding of moss surface at this location is very acidic, but the species is tolerant to these conditions. *Nepenthes spectabilis* and *N. tobaica* are also able to grow in soil with a high humus content or under large tree stands [65]. Soil fertility at the upper montane forest is almost the same as the mossy forest. *Nepenthes* has a very wide adaptability, from soils with low to very high fertility conditions. It grows on soil textures predominately sandy to soils dominated by litter or mosses.

## Population size structure

Knowing the population size structure of a certain plant can help to better understand the population stability and disturbance history [66]. According to our field surveys, the population is dominated by juvenile individuals, followed by mature sterile (without inflorescence) and seedlings. Only a few mature males and females were found during the surveys. The largest populations are found in the upper montane forest, followed by the mossy forest and in open areas on steep slopes near the summit. Very few mature fertile individuals (female and male) and small population sizes implies that these populations are at risk of local extinction. Although the species is able to produce in numerous seeds and grow in several locations in Mount Talang (upper montane forest, mossy forest, and open areas at steep hill near the summit), the population and locations where the species presence need to be protected.

## Prioritizing conservation areas

Determining the priority of conservation areas becomes important to protect the threatened and endemic plant species. In order to set a priority of conservation areas, Species Distribution Models (SDMs) have been widely used to predict the species potential range by relating knowing occurrence to environmental variables where the species is present. Even though SDM models have some problems related to uncertainty and error [67, 68] due to sampling bias and data sparseness [69], the use of SDM in conservation planning becomes a useful tool to select high priority areas for protection [70]. An ensemble model by combining multiple algorithms is widely used to get better predictive performance, previous study stated that ensemble model performs better in predicting suitable areas of endangered giant flower *Amorphophallus titanum* [71]. The ensemble model predicts three suitable classes (suitable, highly suitable and very highly suitable) that are concentrated in the upper montane forest to the summit of Mount Talang. The proposed conservation areas cover 14.9% of total areas of Mount Talang. It is smaller than the unsuitable area of the species (85.1%). The predicted suitable areas can be used as a basis for selecting priority areas that should be protected. The three classes of suitable habitat (suitable, highly suitable and very highly suitable) are located from 1,740 to 2,558 m elevation. According to the actual surveys, the species is observed at 1,819 to 2,489 m elevation. There are still many areas that have not been surveyed, making it difficult to fully evaluate extinction risk. These findings are in accordance with a previous study that found that the species grows in mossy forest and upper montane forest at 1800–2500 m elevation near the summit of mountains [2, 3].

## Future suitable areas under future climate scenario

Global and regional climates are expected to change dramatically during the 21[st] century, [72]. This environmental alteration impacts ecophysiology, distribution, regeneration biology and biotic interaction. The response of plants to environmental changes depends upon functional traits, taxonomy, and life history [73]. Predicted future suitable habitat areas of *N. talangensis* exhibit a small decrease in 2100 using SSP 1–2.6 (a more benign scenario of climate change) and a significant decrease using SSP 5–8.5 (a worst case scenario). The future suitable areas of the species are predicted to shift uphill closer to the summit of the mountain. This is consistent with a previous study that found the climatically-suitable habitat of highland species *Nepenthes tentaculata* is predicted to significantly decrease in 2100 [74]. Montane endemic *Nepenthes* is seriously at risk due to anthropogenic climate change, even a slight change of temperature and precipitation could be quite enough to threaten the species of *Nepenthes*. Climate change could potentially reduce gene flow between the isolated sub-populations, perhaps increasing extinction risks. The best habitat will move upslope isolate sub-population. This pattern has

happened with *N. lowii* and *N. ephippiata* Danser. These species were previously found on Mount Kinabalu and Mount Tambuyukon, but now these are only found in a restricted area of Mount Trusmadi, 55 km to the south [75]. Although some highland Nepenthes are predicted to undergo a decrease of suitable areas in the future climate projections [74, 76], some species may be able to produce a specific metabolite in order to adapt and tolerate to heat and cold stress [77].

## Conservation status assessment

Our assessment showed that *N. talangensis* is qualified for CR under criteria B1ab(iii,v), C2a (ii), and EN under criterion D. Since the species should be listed using the highest category of threat [78], we proposed the category CR B1ab(iii,v), C2a(ii) as the new conservation status of *N. talangensis*. The status has a higher category of threat compared to the current status of the species (EN C2b, ver 2.3). Therefore, *N. talangensis* should be considered to be facing an extremely high risk of extinction in its natural habitat.

## Securing *Nepenthes talangensis* in Botanic Gardens

Seedlings had relatively high survival rate during the five months of observation. The growth was characterized by emergence of new leaves and pitchers. Cuttings also had high survival rate, but the growth was relatively slower. New leaves appeared on some cuttings, but no roots were formed after a few weeks of observation. The cuttings without roots were unable to survive. Further observations are required to better understand growth and reproduction. The use of proper growth media is a critical aspect in securing the species in Botanic Gardens. Several combinations of growth media using moss, cocopeat, and roasted husk show similar numbers of seeds that completely germinating. A previous study reported that mixed media containing moss, cocopeat and roasted husk is a good media for root growth of *N. gracilis*. These media are capable of absorbing sufficient water, have enough Nitrogen (N) and phosphorus (P) availability, and pH in range 5.2–6.41 [79] for plant survival and growth. In terms of securing the species thorough *ex-situ* conservation strategies, it is necessary to consider how the number of individuals that would be taken from nature would affect the sustainability of their populations.

## Conclusions

New occurrence records of *Nepenthes talangensis* have been identified through field surveys. Elevation, canopy cover and slope are variables that best predict suitable habitat for *N. talangensis*. Poor nutrient soils and acidic soils are typical in the locations where the species found. The vegetation composition of sites with *N. talangensis* present is highly variable. The population is dominated by juvenile and mature (sterile) individuals, only a few mature male and female are found in its natural habitat. The surveys included locations where the species is found from 1,819 to 2,489 m elevation, but ensemble modeling predicted that the suitable habitat covers 14.7% of total areas (1,076.5 ha) at 1,740 m to 2,558 m elevation nearby the summit of Mount Talang. The future suitable habitat predicted from climate change scenarios tends to be reduced and at higher elevations. We propose a new conservation status of CR B1ab(iii,v), C2a(ii) based on IUCN Red List Criteria. During attempts to establish seeds and cuttings in Botanic Gardens, some secured individuals were able to live, have a slow growth rate, and produced some new pitchers.

## Supporting information

**S1 Data.**
(CSV)

**S2 Data.**
(CSV)

**S1 File.**
(DOCX)

## Acknowledgments

We would like to thank The Head of Research Center for Plant Conservation, Botanic Gardens and Forestry (BRIN) who supporting us to conduct this study. We also thank to UPTD KPHL Solok for allowing us to conduct field surveys in Mount Talang, West Sumatra, Indonesia. We just wanted to express our appreciation for Harto, a staff of Directorate of Scientific Collection Management who helping us during fieldwork in Mount Talang.

## Author Contributions

**Conceptualization:** Angga Yudaputra, Inggit Puji Astuti, Tri Handayani.

**Data curation:** Angga Yudaputra, Hartutiningsih Siregar, Iyan Robiansyah, Arief Noor Rachmadiyanto, Joko Ridho Witono, Mustaid Siregar, Esti Munawaroh.

**Formal analysis:** Angga Yudaputra, Iyan Robiansyah, Fitriany Amalia Wardhani, Puguh Dwi Raharjo.

**Funding acquisition:** Angga Yudaputra.

**Investigation:** Angga Yudaputra, Tri Handayani, Danang Wahyu Purnomo, Joko Ridho Witono, Izu Andry Fijridiyanto, Yuzammi, Arief Hidayat, Ana Widiana.

**Methodology:** Angga Yudaputra, Fitriany Amalia Wardhani, Puguh Dwi Raharjo, Wendell P. Cropper Jr.

**Project administration:** Angga Yudaputra, Sri Wahyuni, Vandra Kurniawan, Frisca Damayanti, Rizmoon Nurul Zulkarnaen.

**Resources:** Angga Yudaputra, Inggit Puji Astuti, Tri Handayani, Hartutiningsih Siregar, Sri Wahyuni, Arief Noor Rachmadiyanto, Danang Wahyu Purnomo, Vandra Kurniawan, Frisca Damayanti, Rizmoon Nurul Zulkarnaen, Joko Ridho Witono, Mustaid Siregar, Esti Munawaroh.

**Software:** Angga Yudaputra, Iyan Robiansyah, Fitriany Amalia Wardhani, Puguh Dwi Raharjo, Wendell P. Cropper Jr.

**Supervision:** Angga Yudaputra, Inggit Puji Astuti, Tri Handayani, Joko Ridho Witono, Yuzammi, Arief Hidayat, Ana Widiana.

**Validation:** Angga Yudaputra, Izu Andry Fijridiyanto, Wendell P. Cropper Jr.

**Visualization:** Angga Yudaputra, Fitriany Amalia Wardhani, Puguh Dwi Raharjo.

**Writing – original draft:** Angga Yudaputra, Inggit Puji Astuti, Tri Handayani, Hartutiningsih Siregar, Iyan Robiansyah, Sri Wahyuni, Arief Noor Rachmadiyanto, Danang Wahyu Purnomo, Vandra Kurniawan, Frisca Damayanti, Rizmoon Nurul Zulkarnaen, Joko Ridho Witono, Izu Andry Fijridiyanto, Yuzammi,

Arief Hidayat, Mustaid Siregar, Esti Munawaroh, Fitriany Amalia Wardhani, Puguh Dwi Raharjo, Ana Widiana, Wendell P. Cropper Jr.

**Writing – review & editing:** Angga Yudaputra, Inggit Puji Astuti, Tri Handayani, Hartutiningsih Siregar, Iyan Robiansyah, Sri Wahyuni, Arief Noor Rachmadiyanto, Danang Wahyu Purnomo, Vandra Kurniawan, Frisca Damayanti, Rizmoon Nurul Zulkarnaen, Joko Ridho Witono, Izu Andry Fijridiyanto, Yuzammi, Arief Hidayat, Mustaid Siregar, Esti Munawaroh, Fitriany Amalia Wardhani, Puguh Dwi Raharjo, Ana Widiana, Wendell P. Cropper Jr.

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
