## [Decision Letter · Decision Letter 0]

12 May 2023

PONE-D-23-10684Comprehensive approaches for assessing extinction risk of endangered tropical pitcher plant Nepenthes talangensisPLOS ONE

Dear Dr. Yudaputra,

Thank you for submitting your manuscript to PLOS ONE. After careful consideration, we feel that it has merit but does not fully meet PLOS ONE’s publication criteria as it currently stands. Therefore, we invite you to submit a revised version of the manuscript that addresses the points raised during the review process.

We look forward to receiving your revised manuscript.

Kind regards,

Andrea Mastinu

Academic Editor

PLOS ONE

Journal Requirements:

2. We note that Figure (1) in your submission contain [map/satellite] images which may be copyrighted. All PLOS content is published under the Creative Commons Attribution License (CC BY 4.0), which means that the manuscript, images, and Supporting Information files will be freely available online, and any third party is permitted to access, download, copy, distribute, and use these materials in any way, even commercially, with proper attribution. For these reasons, we cannot publish previously copyrighted maps or satellite images created using proprietary data, such as Google software (Google Maps, Street View, and Earth). For more information, see our copyright guidelines: http://journals.plos.org/plosone/s/licenses-and-copyright.

1. You may seek permission from the original copyright holder of Figure(s) [#] to publish the content specifically under the CC BY 4.0 license.  

Natural Earth (public domain): " ext-link-type="uri" xlink:type="simple">http://www.naturalearthdata.com/"

Reviewers' comments:

Reviewer's Responses to Questions

**Comments to the Author**

1. Is the manuscript technically sound, and do the data support the conclusions?

Reviewer #1: No

Reviewer #2: Yes

Reviewer #3: Yes

2. Has the statistical analysis been performed appropriately and rigorously? 

Reviewer #1: N/A

Reviewer #2: N/A

Reviewer #3: Yes

3. Have the authors made all data underlying the findings in their manuscript fully available?

Reviewer #1: Yes

Reviewer #2: Yes

Reviewer #3: Yes

4. Is the manuscript presented in an intelligible fashion and written in standard English?

Reviewer #1: No

Reviewer #2: Yes

Reviewer #3: Yes

5. Review Comments to the Author

Reviewer #1: Although The authors involve Identifying new occurrence records and characterizing ecological requirements, Population size structure, Prioritizing conservation areas, Future suitable areas under future climate scenarios, Conservation status assessment, and Securing individuals at Botanic Gardens during their study intitled Comprehensive approaches for assessing extinction risk of endangered tropical pitcher plant Nepenthes talangensis, the authors overlooked some important details

Reviewer #2: The manuscript presents a valuable contribution to the conservation of Nepenthes talangensis, a species at risk due to climate change. While the study identifies the potential loss of suitable habitat for the species under various climate change scenarios, it falls short in providing practical solutions for mitigating this threat. However, the authors' findings on the germination and growth of N. talangensis under different media provide useful insights into its cultivation in botanic gardens, which can help in its conservation efforts. The authors' detailed observations on the survival and growth of seedlings and cuttings are commendable. Overall, the manuscript offers valuable insights into the potential impact of climate change on N. talangensis and highlights the need for continued conservation efforts to protect this species.

Reviewer #3: Please see attachment for specific comments.

I think the CSVs in the SI could have a line on top of the data explaining what the data are.

I think the statistical analysis is sound, but you never mentioned in the paper where the 'absence' data came from, this would be helpful!

6. PLOS authors have the option to publish the peer review history of their article (what does this mean?). If published, this will include your full peer review and any attached files.

Reviewer #1: No

Reviewer #2: No

Reviewer #3: No

---

## [Author Response · Author response to Decision Letter 0]

14 Jun 2023

We have responded comments from the reviewers. The required revision files include revised manuscript, revision with track changes, response to reviewer, revised figure and rebuttal letters have been attached in this system

---

## [Decision Letter · Decision Letter 1]

25 Jul 2023

Comprehensive approaches for assessing extinction risk of endangered tropical pitcher plant Nepenthes talangensis

PONE-D-23-10684R1

Dear Dr. Yudaputra,

We’re pleased to inform you that your manuscript has been judged scientifically suitable for publication and will be formally accepted for publication once it meets all outstanding technical requirements.

Kind regards,

Andrea Mastinu

Academic Editor

PLOS ONE

Additional Editor Comments (optional):

Reviewers' comments:

Reviewer's Responses to Questions

**Comments to the Author**

1. If the authors have adequately addressed your comments raised in a previous round of review and you feel that this manuscript is now acceptable for publication, you may indicate that here to bypass the “Comments to the Author” section, enter your conflict of interest statement in the “Confidential to Editor” section, and submit your "Accept" recommendation.

Reviewer #2: All comments have been addressed

2. Is the manuscript technically sound, and do the data support the conclusions?

Reviewer #2: Yes

3. Has the statistical analysis been performed appropriately and rigorously? 

Reviewer #2: Yes

4. Have the authors made all data underlying the findings in their manuscript fully available?

Reviewer #2: Yes

5. Is the manuscript presented in an intelligible fashion and written in standard English?

Reviewer #2: Yes

6. Review Comments to the Author

Reviewer #2: Dear authors,

The vast majority of comments have been addressed, the manuscript can be published. Thank you for all the hard work for this threatened microendemic. Well done!

7. PLOS authors have the option to publish the peer review history of their article (what does this mean?). If published, this will include your full peer review and any attached files.

Reviewer #2: No

---

## [Editor Report · Acceptance letter]

28 Jul 2023

PONE-D-23-10684R1 

Comprehensive approaches for assessing extinction risk of endangered tropical pitcher plant *Nepenthes talangensis*

Dear Dr. Yudaputra:

I'm pleased to inform you that your manuscript has been deemed suitable for publication in PLOS ONE. Congratulations! Your manuscript is now with our production department. 

Kind regards, 

on behalf of

Dr. Andrea Mastinu 

Academic Editor

PLOS ONE